# "My drinking was way worse during the pandemic": A qualitative analysis of contextual and individual factors impacting alcohol use during the COVID-19 pandemic

Anthony Surace[1,2*], Cat Munroe[2,3], Priscilla Martinez[2]

1 School of Public Health, University of California Berkeley, Berkeley California, United States of America,
2 Alcohol Research Group, Public Health Institute, Emeryville, California, United States of America,
3 Department of Psychology, Purchase College SUNY, Purchase, United States of America

* asurace@berkeley.edu

## Abstract

### Background

Evidence shows that alcohol use in the United States increased during the COVID-19 pandemic. This primarily quantitative work has not examined how the unique context of the COVID-19 pandemic in the U.S. may have shaped motivations for alcohol use during this crisis. To address this knowledge gap, we conducted an analysis of qualitative data from in-depth interviews examining people's motivations for using alcohol during the COVID-19 pandemic.

### Methods

Participants (N=26) were derived from those who completed all three waves of the National Alcohol Survey COVID Cohort- a longitudinal population-based survey of non-institutionalized U.S. adults. Interviews were conducted from April-July 2022 over Zoom. Interview transcripts were analyzed iteratively via codebook thematic analysis.

### Results

Participants described how both contextual and individual level factors resulted in increased alcohol use. Contextual factors included reductions in barriers to alcohol use and increases in alcohol availability and accessibility. Individual level factors included using alcohol to regulate emotions (e.g., to alleviate boredom and as an end of day "reward") and to celebrate reconnecting with loved ones once social distancing restrictions began to ease.

### Conclusions

Our results suggest that environmental forces may have interacted with individuals' emotions to shape alcohol use motivations during the COVID-19 pandemic. This work helps to contextualize quantitative research on changes in alcohol use observed during

**Data availability statement:** Data cannot be shared publicly because of concerns about participant confidentiality. That is, participants could potentially be identified via a combination of the small number of participants, analyzed demographic information in the paper, and personal details in the interview transcripts. Data are available from the Alcohol Research Group (contact via info@arg.org) for researchers who meet the criteria for access to confidential data.

**Funding:** This work was funded by the National Institute of Alcohol Abuse and Alcoholism at the National Institutes of Health (T32AA007240, WK and R01AA029921, PM). Opinions are those of the authors and may not represent official positions of the National Institute of Alcohol Abuse and Alcoholism or the National Institutes of Health, which had no role in the analysis or interpretation of the data.

**Competing interests:** The authors have declared that no competing interests exist.

the COVID-19 pandemic. More research is needed to determine the long-term impacts of pandemic era changes in alcohol use. It is also necessary for future studies to examine how such impacts may manifest differently across the U.S. population (e.g., among racial/ethnic minority individuals).

## Introduction

The United States (U.S.) is one of the few high-income countries that experienced an average increase in alcohol use during the COVID-19 pandemic [1,2]. Cross-sectional, retrospective studies conducted early in the pandemic showed increases in drinking frequency [3], drinks per month [4,5], drinking days [6], and binge drinking [7], and longitudinal cohort studies supported these observations [2,8,9]. For example, data from the National Alcohol Survey COVID Cohort (NAS-C19) showed increases in alcohol use and symptoms of alcohol use disorder in the first year of the pandemic, driven by an increase in spirits use and frequency of drinking rather than increases in average quantity per occasion [2]. Additionally, per capita alcohol consumption increased by 5.5% from 2019 to 2021, the largest two-year increase since 1969 [10]. Increases in alcohol use during the pandemic, however, varied across populations [2], with women [11], Black people [2,7], people with mental health problems [12,13], and those engaging in heavy alcohol use before the pandemic [14–16] being especially likely to report increases in drinking.

These increases align with previous research showing higher alcohol use after exposure to natural disasters and human-caused crises [17–19]. In the U.S., people exposed to trauma from the September 11, 2001 terrorist attacks and first responders to Hurricane Katrina increased their alcohol use, some up to years afterward [20–23]. The psychosocial and behavioral mechanisms through which these disasters impacted alcohol use include symptoms of Post-Traumatic Stress Disorder [24] and reduced social cohesion [25]. The COVID-19 pandemic, however, differed from other modern-day disasters, and thus may have impacted alcohol use both through previously identified mechanisms and those unique to an infectious disease pandemic. Unlike previous crises, the COVID-19 pandemic required people to restrict social interactions (e.g., stay-at-home orders, self-quarantines, etc.) for prolonged periods. Reviews of the psychological impacts of quarantine report that protracted social isolation due to quarantining was associated with feelings of confusion and boredom, and negative mental health effects including anxiety and depression [26,27]. Important for understanding reasons for using alcohol during the pandemic in the U.S., research among U.S. respondents shows that social isolation and loneliness resulting from COVID-19 quarantines were associated with increased alcohol use during the pandemic [28].

Quantitative investigations designed to understand reasons for increased alcohol use in the U.S. during the COVID-19 pandemic have identified economic [12], family [29], and mental health stressors [12] as drivers of increased alcohol use. In addition, other motives related to the COVID-19 pandemic have been identified. For example, a study using national Monitoring the Future data to examine trends in alcohol use and drinking context and reasons among young and middle-aged adults observed that increases in alcohol use were associated with motives to relax/relieve tension and alleviate boredom resultant of the pandemic, for example due to stay-at-home directives [3]. Similarly, a study using National Alcohol Survey data observed that reports of drinking to forget one's worries and problems were higher in the early-COVID-19 period relative to the period just prior [11]. A number of studies identified associations between drinking to cope and increased alcohol use during the COVID-19 pandemic among a variety of populations including residents of Harlem, New York City [30],

college students [31,32], women [33], women who parent [34], and members of the LGBTQ+ community [35,36]. Although such research is vital for identifying factors associated with observed increases in alcohol use during the pandemic, they are of limited use in elucidating participants' rationale for drinking. That is, unlike qualitative data, quantitative data cannot tell us how people perceived the ways that U.S.-specific environmental forces unique to the COVID-19 pandemic interacted with their affective states to shape thought processes and motivations for alcohol use during this global crisis [37].

Relative to quantitative studies, limited qualitative research to date has examined motives for alcohol use during the COVID-19 pandemic in the U.S. For example, Finlay et. al. examined motives for alcohol use among older adults and observed that those who increased their alcohol consumption did so to relax, enjoy an evening routine, or "get drunk" [38]. A descriptive phenomenological study of sexual minority women (e.g., self-identified lesbians) reported that changes in routine, and seeking recreation and relief were associated with alcohol and cannabis use [39]. An online study of U.S. veterans who reported substance use issues during the COVID-19 pandemic used a single item asking the participant to "describe anything else related to the impact of coronavirus on you, such as your use of substances/alcohol, mood, relationships, or health" [40]. Narrative responses were examined using a modified consensual qualitative approach and found that increased psychological distress, reduced access to recreational options, feelings of loneliness/isolation and relationship issues impacted participants' substance use. A follow-up to the Eating and Activity over Time cohort study among young people in Minneapolis-St. Paul also used open-ended items to assess experiences of stress, stress management, and substance use among participants during the COVID-19 pandemic. Using an inductive approach to thematic analysis the study reported that participants engaged in alcohol use as a strategy for managing stress during the pandemic [41].

Taken together, qualitative literature suggests a variety of motives for using alcohol during the COVID-19 pandemic, ranging from wanting to relax to coping with negative feelings resulting from living through a global crisis. However, the limited number of studies and use of narrative, written responses to open-ended questions on surveys are a limitation. Written survey responses are not analogous to in-depth conversations about the motives and circumstances for using alcohol during the pandemic. Open-ended survey question responses can be valuable data, but they lack "richness" [42]- they are often short and lack information on the social, contextual, and/or personal factors that impact behavior. Further, the nature of surveys precludes follow-up questions that would contextualize or clarify responses. Data derived from such methods are of limited use in drawing inferences about the lived experiences and motivations for using alcohol during a pandemic. Understanding the nuanced and in-depth aspects of people's reasons for using alcohol during a global pandemic can inform concerted responses to support people during crises in the future.

To address this knowledge gap, we analyzed qualitative data from in-depth interviews with participants of the NAS-C19 [2]. We aimed to describe the range of motivations for using alcohol during the COVID-19 pandemic among U.S.-based individuals.

## Methods

### Sample

This qualitative study (N = 26) drew participants from the NAS-C19, a longitudinal population-based survey of non-institutionalized adults aged 18 or older across all 50 states and the District of Columbia. The NAS-C19 included three waves of data collection: 2019-2020 (Wave 1, pre-pandemic), in 2021 (Wave 2), and in 2022 (Wave 3). Participants completed measures of alcohol use and alcohol use disorder (AUD) symptomology at each survey via

items derived from the Diagnostic and Statistical Manual – Fifth Edition (DSM-V; [43]). Participants in Wave 1 were sampled using both probability-based (random digit dialing with computer assisted telephone interviews and address-based sampling via an online survey) and non-probability-based (web panel) approaches and included oversamples of Black and Hispanic/Latinx people. For more details on NAS-C19 procedures, see Kerr et al. [2]. Eligibility criteria for the current subsample included 1) completion of all survey waves, and 2) self-reported past-year drinking above National Institute on Alcohol Abuse and Alcoholism (NIAAA) guidelines: that is, > 5 drinks/day or > 15 drinks/week for men; > 4 drinks/day or > 8 drinks/week for women [44] at Wave 2.

## Procedures

Eligible NAS-C19 participants were invited to participate in the present study at the end of their Wave 3 survey. Wave 3 of the survey was conducted from March 16th - June 21st, 2022. Interested participants were contacted up to three times via email about participation. Participants were informed about the interview topic and procedures, scheduling logistics, and the gift card they would receive after participation. Interested respondents were scheduled for an hour-and-a-half-long study appointment over Zoom. The response rate for the qualitative interviews was 29.5%.

All study interviews occurred between April 11th and July 6th of 2022. Before beginning the interviews, interviewers gave an overview of the study's purpose and obtained verbal consent for participation. Then, the interviewer conducted a semi-structured interview using an interview guide with a priori questions. All participants were asked the same questions (e.g., thoughts and feelings about using alcohol during the pandemic), but interviewers had the flexibility to explore topics organically as they arose during the interview.

Participants described their drinking over the course of the prior year, and the participant and interviewer worked together to note major life events in the participant's life that occurred during the prior 12 months to aid participants' recall of their alcohol use during that time frame. Participants were asked about stressful experiences that occurred, when they were drinking the most/least, how they understood these periods of drinking the most/least (i.e., what thoughts and feelings they had about these periods, motivations for using alcohol or not), and how their alcohol use during the period compared to their alcohol use pre-pandemic. Significant life events were probed for their associations with alcohol use, as were perceptions of how alcohol use may have affected later life events. After the interview, the first 22 participants were sent a gift card for $50. Towards the end of the study, the incentive was increased to improve recruitment resulting in the last four participants receiving a $75 gift card.

All procedures were approved by the Public Health Institute institutional review board.

## Analysis

All interviews were professionally transcribed. Transcripts were then analyzed using codebook thematic analysis, an iterative technique in which initial broad research questions inform the abductive generation of themes [45]. When using thematic analysis, initial primary (open) codes are developed, and then connected to other related themes to form overarching secondary codes that are developed into themes.

Our analysis occurred as follows: First, authors one and two independently performed open coding of two full transcripts. Emerging themes were discussed between the two primary coders and author three until consensus was reached. Next, authors one and two independently coded additional transcripts, discussing emerging themes, coding agreement, and

iteratively developing the codebook (managed by author one). When coding a new transcript did not result in identification of potential new themes, the codebook was determined to be complete and authors one and two reviewed their coding of early transcripts to confirm all codes in the final codebook were applied as relevant, and to determine agreement between the two coders. Throughout analysis, authors one, two, and three met weekly to discuss coding and address discrepancies until all transcripts were coded. All transcripts were double-coded.

## Results

Sample demographics are available in Table 1. Participants' self-reported alcohol use behavior and alcohol use disorder symptomology (both pre- and post-COVID) are available in Table 2. Our analysis revealed that participants' alcohol use was shaped by both contextual and individual-level factors (for a summary of identified themes and exemplary quotes see Supplementary Table 1).

**Table 1. Participant Demographics.**

| Characteristics | Total ($N = 26$) |
| --- | --- |
| | *Mean* (SD) or *n* (%) |
| Age (Range: 23 – 66) | 39.38 (12.7) |
| Gender | |
| Female | 16(61.5%) |
| Male | 10(38.5%) |
| Race/Ethnicity | |
| Hispanic | 8(30.8%) |
| White | 12(46.2%) |
| Black or African American | 5(19.2%) |
| Other | 1(3.9%) |
| Marital Status | |
| Married/co-habitating | 12(46.2%) |
| Divorced | 2(7.7%) |
| Never married | 12(46.2%) |
| Education | |
| High School | 1(3.9%) |
| Some college | 4(15.4%) |
| College or more | 21(80.8%) |
| Household Income | |
| $70,000 or more | 14(53.9%) |
| Employed (Pre-COVID) | 21(80.8%) |
| Lost Employment during COVID | 7(26.9%) |
| Essential Worker during COVID | 14(53.85%) |
| Geographic location | |
| Northeast | 4(15.4%) |
| Midwest | 4(15.4%) |
| Pacific | 3(11.5%) |
| South | 10(38.5%) |
| Mountain | 5(19.2%) |
| Average number of children per household | 1(1.7) |

## Contextual factors

Analysis of interview transcripts revealed that the social and environmental conditions created by the COVID-19 pandemic led to participants experiencing multiple facilitators of alcohol use which resulted in participants' alcohol use increasing relative to pre-pandemic. Three such facilitators included 1) a reduction in barriers to alcohol use, 2) the increased availability of alcohol (i.e., the ease with which participants could *acquire* alcohol), and 3) the increased accessibility of alcohol (i.e., the ease with which participants could *consume* alcohol).

### Theme 1: During the COVID-19 pandemic, participants encountered fewer barriers to alcohol use resulting in changes to their drinking patterns

Participants reported that an effect of stay-at-home orders and restrictions on movement during the COVID-19 pandemic was a reduction in barriers to drinking. That is, factors which would usually present an obstacle to drinking pre-pandemic no longer applied during the COVID-19 pandemic. Such obstacles could be physical (e.g., having to travel to a bar to drink alcohol) or social (e.g., normative beliefs about when alcohol should be consumed). The removal of such obstacles resulted in some participants drinking more during the pandemic than they had previously. As one participant articulated, "I think… because (alcohol use) was like less limited by other factors I think we were just drinking more than usual" (PM_15, 24-year-old non-Hispanic White female with moderate AUD symptomology). This sentiment was elaborated on by another participant who stated that:

> I didn't have to go into the office. There wasn't as much stuff going on, especially when they didn't have like sports or anything. My kids have like three activities a day. So, to go from like 100% to like nothing, I was like "Wow, nothing else to do," (PM_11, 37-year-old non-Hispanic, White female with mild AUD symptomology).

Factors that normally served as deterrents to alcohol use (e.g., time constraints) tended to fall within two categories. The first was restrictions on participants' ambulatory ability (e.g., "stay-at-home orders"; [46]) which led some participants to report drinking more than they normally would. For example, one participant said:

**Table 2. Participant Alcohol Use Disorder Symptomology and Alcohol Use Behavior.**

|  | Pre-COVID Total (*N* = 26) *Mean* (SD) or *n* (%) | Post-COVID Total (*N* = 26) *Mean* (SD) or *n* (%) | T/z | *p* |
|---|---|---|---|---|
| Alcohol Use Disorder Symptomology Scores* |  |  |  |  |
| No AUD (0-1) | 15(57.7%) | 10(38.46%) | -2.4 | .02 |
| Mild AUD (2-3) | 5(19.2%) | 7(26.9%) |  |  |
| Moderate AUD (4-5) | 3(11.5%) | 3(11.5%) |  |  |
| Severe AUD (6+) | 3(11.5%) | 6(23.1%) |  |  |
| Self-Reported Alcohol Use |  |  |  |  |
| Total weekly volume (standard drinks) | 9.3(12.5) | 13.7(16.3) | -2.2 | .04 |
| Frequency of heavy alcohol use** | 38.9(73.0) | 69(94.6) | -1.9 | .04 |
| Drinking at home at least monthly | 16(66.7%) | 22(84.62%) | 2.2 | .04 |
| Drinking outside the home (bars) at least monthly | 14(53.9%) | 8(30.8%) | -2.9 | .00 |

*Alcohol use disorder symptoms was determined via items derived from the DSM-V(46).

**4+/5+ standard drinks in a sitting for women and men respectively

> I think I (drank) more often primarily. But as far as like quantities go too like again when you go out to the bars, you don't want to get like super drunk because you want to make sure you make it home safe. I've always been super responsible about that. But if I'm already at home and I'm sleeping in the same place that I'm drinking, I feel really comfortable drinking more. And we had-- like my birthday that year we had like a-- we threw like a little disco. We got like a disco ball for our living room. And we got like a bunch of booze and we had maybe like six or seven people over and it was really fun. But like I said I could drink more because I'm at home, whereas the previous year we went out to eat or something like that. I don't know. So it was definitely increased. (PM_15, 24-year-old non-Hispanic White female with moderate AUD symptomology)

In this instance, the participant describes how pre-pandemic, they would use alcohol outside of their home in public settings (i.e., at bars). The need to safely return to their residence after using alcohol resulted in the subject moderating the amount of alcohol they would consume. Due to restrictions on public gatherings during the COVID-19 pandemic, however, this participant was unable to drink outside their home. This eliminated a pre-pandemic moderator of drinking behavior, resulting in the participant reporting using more alcohol. These sentiments were echoed by another participant who declared: "We were used to going out. So I think we still drink as much, but if not more, because now we're home, we're not out somewhere" (PM_06, 36-year-old Hispanic White female with severe AUD symptomology).

In a similar vein, changes in work-related routines diminished the usual physical or social barriers participants would typically experience to using alcohol. For instance, due to unemployment or reductions in work obligations during the pandemic, participants experienced fewer obstacles to drinking. As one participant described:

> I was home. I couldn't go to the gym. Work had basically ceased because no one could really do anything. I had my cell phone and now, I had-- all my friends like beer and drink or most of them do, a lot of my family does, and the guys at work that I work with are all really into beer. So, for those initial few weeks in March after lockdown, the reality was people were drinking beer at lunch and sending pictures of it around and I'm sitting on my couch at noon on my lunch break and I have no kids, no dependents. It's just me and I'm like "That beer looks pretty good. I'm going to have a beer." So, I started doing that and I wouldn't say my drinking got crazy, but it was definitely more than what my average was… (CM_09, 51-year-old non-Hispanic White male with no AUD symptomology)

The previous quote also showcases how social norms around drinking may have shifted during the COVID-19 pandemic. In this case, due to a lack of work obligations, the participant and his colleagues began drinking during their lunch breaks and sharing pictures of their beer with each other, socially reinforcing their shared change in alcohol use. Another participant who described drinking during the pandemic stated:

> …everyone else was bored. So, [my friends would] be like "Oh, I'm going to come over and sit on the porch and drink with you," and I'm like "Great. That's what we're going to do. Nothing else to do." (PM_11, 37-year-old non-Hispanic, White female with severe AUD symptomology)

In this case, the participant and their peers shared the experience of having fewer obligations and thus reported engaging in alcohol use. Even among participants whose work schedules were relatively unaffected by COVID-19 restrictions, changes in drinking behavior were apparent. For example, one participant said:

> …most of the time since I work from home I don't have to leave except for going to get groceries and stuff, I just pretty much stayed here all the time drinking because there was no reason for me to leave... (PM_03, 27-year-old non-Hispanic Multiracial male with severe AUD symptomology)

Similarly, another participant stated

> …it was easier just to start drinking while working at the house. There's nobody there supervising me. So I got into that bad habit. I would be drinking while working from home… that was new for me since I would never drink and work at the same time. (CM_11, 33-year-old Hispanic White female with severe AUD symptomology)

Taken together, these quotes suggest that for some participants working from home motivated an increase in alcohol use. This seems to have been due to the removal of previous drinking barriers including physical limitations (e.g., driving or travelling to bars). Additionally, it is evident that the social norms around alcohol use shifted because of social distancing measures such that, for example, drinking at certain times during the day became socially acceptable when previously it was not. In addition to these social barriers, there is evidence to suggest that changes to the availability of alcohol also facilitated reported increases in alcohol use during the COVID-19 pandemic.

### Theme 2: During the COVID-19 pandemic, increased availability of alcohol facilitated alcohol use

Participants reported that the proliferation of options for purchasing alcohol made it easier for them to drink during the COVID-19 pandemic. That is, the environmental conditions dictating the degree to which the participants could get alcohol was a contributing factor to its use among participants. This availability could take on many forms. For example, numerous participants noted the expansions in local alcohol delivery including alcohol pick-up and to-go drink options. Respondents noted that they likely would not have utilized alcohol delivery pre-pandemic, but the convenience of this option was very appealing during the pandemic when social distancing measures were being promoted. For instance, one interaction between an interviewer and participant was as follows:

> Participant: Oh, yeah. My drinking was way worse during the pandemic than it is now.
>
> Interviewer: Yeah. How do you understand that? What's contributed...
>
> Participant: Well, it was more available because I know here in [southwestern state], the Governor had approved for alcohol to be delivered to the house. Like, if you would order it over the phone, it could be delivered during COVID. So, I would utilize that sometimes. (CM_11, 33-year-old Hispanic White female with severe AUD symptomology)

This is echoed by another participant exchange

> Interviewer: Was it easy or hard to get alcohol during the pandemic for you? Or was it about the same?
>
> Participant: It was easier. I mean, you have so many places, like, selling to go slushies in a bag from a bar. <laughs> Like, I mean, come on, now. (CM_07, 38-year-old non-Hispanic White female with moderate AUD symptomology)

Still another participant reported

> But yeah, I think the easy access, you know, the to-go thing, the outdoor venues that you
> can drink outside it felt like kind of-- some of the previous boundaries have softened. It felt
> a bit more like Europe because I'm like-- we grow up and go in public transportation have
> drinks with us. That was normal or it still would be normal, but it feels strange. If I go back
> and I'll go, like, what are you guys doing? What? (PM_12, 47-year-old non-Hispanic White
> male with mild AUD symptomology)

These statements suggest that changes in alcohol availability (i.e., alcohol delivery and to-go
policies) played a direct role in motivating alcohol use during the COVID-19 pandemic. In
these cases, the expansion of alcohol delivery and to-go policies facilitated alcohol use among
participants. This seems to have been compounded by alcohol already being endemically
prevalent. As one participant described:

> Interviewer: Why do you think it was alcohol and maybe not something else?
>
> Participant: …There's a liquor store almost everywhere. There's a beer store almost every-
> where. It's easy to get and you can even order it for delivery as well. So, I think that's the
> reason why it became alcohol and nothing else and because it's legal too. So, I can't smoke
> weed because of all this other stuff, but I know a lot of folks who use substances like that,
> but alcohol just became something that was available and easy to get. (PM_02, 36-year-old
> non-Hispanic Black male with no AUD symptomology)

In this example, the participant describes how since alcohol is legal and liquor stores are com-
monplace, it was easy for them to drink. This high availability of alcohol allowed participants
to readily acquire alcohol and consume it. In addition, greater availability via the expansion
of alcohol delivery services and to-go drinks from on-premises alcohol outlets led to partici-
pants having more options for alcohol availability which also supported alcohol use during the
COVID-19 pandemic.

## Theme 3: During the COVID-19 pandemic, alcohol became more easily accessible which fostered increased alcohol use

Participants reported how the ease with which they could consume alcohol was a contributing
factor to changes in their alcohol use during the COVID-19 pandemic. This alcohol accessibil-
ity was defined as the environmental conditions dictating how quickly/easily the participant
could consume alcohol. For example, always having beer in the house making alcohol imme-
diately accessible. Participants reported that having alcohol in their homes was a contributing
factor to their alcohol use. For example, one participant described:

> Yeah, no it changed like the convenience and right there. And it's sort of-- I was looking for
> an outlet, and that was the easy hook for me. Okay, go to the fridge, get it. And I know over
> time if I'm looking at this, like I would start craving when is this thing over so I can go get
> this thing. (PM_02, 36-year-old non-Hispanic Black female with no AUD symptomology)

For context, this participant was discussing how they utilized alcohol to minimize the stress
they experienced from their job during the COIVID-19 pandemic. In this case, the imme-
diate accessibility of alcohol facilitated them utilizing it as a stress reliever. When asked why
they used alcohol instead of another strategy to relieve stress this participant went on to say

"I don't know. I think it's because it was there, the power of suggestion". Taken together these statements suggest that for some participants during the COVID-19 pandemic, the accessibility of alcohol facilitated its use as a stress reliever more so than before and in lieu of alternatives. Had there not been COVID-19 related restrictions (e.g., stay at home orders, social distancing measures) then this person may have reported using other means to relieve stress (e.g., going to the gym). Another participant said something similar:

> I think just like the accessibility of [alcohol], knowing like "Oh, let me just walk two feet, pour myself my own drink. There's no bar tab that I can see," and not having to like get up in the morning, get the kids to school or when they were in school, take them to school and then I could just go back home. I didn't have to go into the office. There wasn't as much stuff going on (PM_11, 37-year-old non-Hispanic White female with severe AUD symptomology)

### Another participant stated

> We can drink as much as we want. And I know the wine rack was always full during those early parts of the pandemic, which was great, but also, I definitely was drinking more. (PM_15, 24-year-old non-Hispanic White female with no AUD symptomology)

The above quotes suggest that for these participants the fact that alcohol was readily accessible (i.e., it was in their houses) was a facilitator of their use of alcohol. Considering these participant statements, it becomes clear that alcohol accessibility had an impact on changes in alcohol use behavior during the COVID-19 pandemic.

### Personal factors

Participants noted multiple personal reasons for drinking in the context of the COVID-19 pandemic. These included emotion-focused reasons, such as 1) coping with negative feelings that resulted from being unmoored and untethered from their everyday life due to the COVID-19 landscape; stress, anxiety, and emotional malaise; and boredom imposed by shelter-in-place; 2) the desire to experience positive emotion and pleasure in the context of ongoing deprivation; and 3) in order to celebrate (re)connections with loved ones and social events once social distancing restrictions began to ease.

### Theme 4: Participants described drinking to relax, unwind, and escape chronic negative emotions during the COVID-19 pandemic

Participants described drinking alcohol to find relief from the negative affect resultant of the COVID-19 pandemic. Negative emotions experienced by participants included boredom (e.g., due to social gathering restrictions), stress due to changing work conditions during the pandemic, and social isolation. Participants described these negative emotions as the product of specific pandemic-related stressors (e.g., schooling children from home). For example, one participant stated:

> In many ways (drinking) was to relax. You know, being able to put the children to bed and think, okay, I've got at least an hour before somebody cries or something. I'm going got (sic) watch, you know, half an hour show. And then, you know, sit down on the couch and then… we would have that because we liked it, and then it would be in the fridge, and

it would be so easy to just grab it and open it and go. And so it was an easeability (sic). It was, you know, what I would consider like a quick relaxation type of a situation. (PM_08, 31-year-old non-Hispanic White female with no AUD symptomology)

In this case the participant described how alcohol was used to relax during the COVID-19 pandemic. For context, the participant had described earlier in the interview how changes in their job role and responsibilities during the COVID-19 pandemic had disrupted their child-care. Additionally, the participant described how the accessibility of alcohol was what made it such an attractive option for stress relief. This accessibility seems to have been unique to the COVID-19 pandemic: no participants described using alcohol for relieving stress because it was always accessible prior to the COVID-19 pandemic.

In addition, participants described an indefinable feeling of malaise/boredom during the pandemic. Often, participants described drinking to temporarily separate themselves from such feelings. Related to boredom specifically, one participant responded to a question about their reasons for drinking by saying, "I don't know, I feel like a lot of drinking, especially during COVID, was just out of boredom, like being home and being bored" (PM_11, 37-year-old non-Hispanic White female with severe AUD symptomology). Another participant described her experience of increasing her drinking during the pandemic in the following way:

I felt like I was on vacation, but it wasn't a happy vacation…when I started drinking my heaviest I was just kind of like, well, I don't have to go anywhere. Well, I don't care. Well, I had a rough day. I'm going to do it. And so it was kind of like this like I'm in a really bad place…. I wasn't like drinking a bottle of wine a day type of a situation, but I would have a beer while I was like unwinding watching a show at the end of the day. And I didn't think about it at all. I just did it during that time. (PM_08, 31-year-old non-Hispanic White female with no AUD symptomology)

She went on to note that drinking alcohol provided a way of creating a sense of perceived control during the pandemic, when she felt little control over the chronic stress in her life:

Well, it definitely helped me to relax. I think on this weird level there was kind of like this feeling that I didn't have control over anything… I can go to the liquor store, and I can pick out my own beer. And, you know, I got everything else taken out from under me, but I can do that. And then I can have a drink on a Monday night and nobody's going to tell me I can't. So it was bad, I think. (PM 08, 31-year-old non-Hispanic White female with no AUD symptomology)

Other participants directly tied increases in their drinking to increases in stress at work, particularly among those who were essential workers and had to work in-person during the pandemic. One participant who worked in healthcare shared:

I was noticeably drinking more and especially because things were stressful working in healthcare during those times. Our staffing was crummy…after a hard day you want a glass of wine with dinner to wind down and it wasn't usually more than a glass or two, but it was like almost at--it was like five or six days a week. So if I opened a bottle last night and had two glasses, then the next day, I had to finish the bottle with another two glasses. And then it was kind of--it wasn't like that every single week, but it was definitely more frequent and noticeable. (PM_15, 24-year-old non-Hispanic White female with moderate AUD symptomology)

In these instances, participants described stressors associated with the pandemic, including feelings of being out of control, in tandem with decreased opportunities to regulate or escape their stressful contexts, as being associated with increases in their alcohol use.

Some participants also tied their alcohol use during the pandemic to social isolation. As one participant described:

> I probably was drinking more during that time. I had a very strange schedule at work. I do work out in the environment, so there would be times where I would be just in a hotel room for a week by myself, so kind of a unique situation which probably culminated in I'm by myself more so I'm drinking, I don't really have… I was alone a lot and there was really nothing to do where I was besides drink if that's what I-- if I didn't fall asleep right away. (CM_08 33-year-old non-Hispanic White female with mild AUD symptomology)

This participant was in a unique position where their job required them to travel during the pandemic. However, due to social distancing measures they were essentially quarantined in their hotel room while not at their jobsite. This person described how this resulted in being socially isolated and thus drank more during the pandemic. Similarly, another participant described how

> …I want to say yeah, 2020, we at least-- well, of course, I was working at the bank and taking every precaution. Nobody there at work had-- it just seemed like it was very distant, you know, even though in [Southwestern City], the area was getting hit hard, it's just in my circle of who I knew and where I was all the time, it just was very distant. So, Christmas season… it was just like total isolation and none of the regular stuff. So, I guess that would be the main stressor of it didn't even really feel like Christmas because just... There was no family at any social-- when it came to family. …I guess [I was drinking] every weekend. (PM_01 30-year-old Hispanic male with mild AUD symptomology)

In this instance, the participant describes how during the COVID-19 pandemic they experienced "total isolation". This participant was socially isolated from their peers at work and was unable to spend the Christmas holidays with their family due to COVID-19. Concurrently they described their alcohol use as increasing. In both quotes above participants described how the social isolation resultant of the COIVD-19 pandemic was associated with changes in their alcohol use.

### Theme 5: Participants described drinking to experience positive emotions in a context with reduced opportunities for rewarding activities

Some participants described using alcohol to provide themselves with a reward or to feel positive emotions in contexts of ongoing affective deprivation. Participants described the COVID-19 pandemic as an environment with diminished opportunities for reward, positive emotion, and pleasure. In that context, participants described using alcohol simply because they missed experiencing positive emotions and had few other ways to access pleasure. One participant shared:

> I think in a positive light, it's been a way to feel like I can still have a treat…I can't go out to restaurants, I can't see friends, but it's like you know what, I can have a glass of wine and I can watch a movie, and that is a nice way to feel like I'm taking care of myself…It's just like it's something nice I feel like I can do for myself that's relatively low impact. (PM_13, 25-year-old non-Hispanic White female with no AUD symptomology).

In a similar vein, a different participant stated the following:

> I think it's purely a pleasure thing… not to get too graphic, but with COVID a girl that I was dating, we split up during COVID-19. She lives like 40 minutes for me. And I just-- you know, I wasn't comfortable like going-- and she's got kids and everything and so we dissolved that. So it-- there's like less pleasure in my life now than there was pre-COVID. And one of the things that I can do is, you know, pour an [brand name] beer or, you know, a nice Oktoberfest and for that half hour or 45 minutes, I can, you know, I can enjoy it. (CM_09, 51-year-old non-Hispanic White male with no AUD symptomology)

In this instance the participant describes how the COVID-19 resulted in a "lack of pleasure" in their life. This participant's romantic relationship ended due to the pandemic and previously in the interview they had described how social gathering restrictions had resulted in a lack of alternative means of stimulation (e.g., not being able to go to the gym). As such, for this person alcohol became a source of positive reinforcement. Another participant also specifically described drinking to reward himself for getting through the work week or as a reward after working out, saying "During COVID, the beer became a primary focus. I sat on my couch and really focused on every sip because it was a sensual experience that I wasn't getting anywhere else." (CM_06, 54-year-old non-Hispanic White male with no AUD symptomology). In these examples, participants describe alcohol as creating a positive emotional and sensory experience, and as a way of being able to access pleasure and positive emotion.

### Theme 6: As pandemic lockdowns began to lift, participants described drinking more heavily during social events and drinking alcohol to celebrate (renewed) ability to spend time with friends and family

Multiple participants noted that the easing of COVID-19 related social distancing measures motivated heavy alcohol use. Specifically, as they began to re-engage with friends and family whom they had not been able to connect with during the pandemic they drank more during these social interactions than they had pre-pandemic. Participants described how due to overall feelings of deprivation during the COVID-19 pandemic, social events that were normative pre-pandemic felt particularly celebratory. As a result, they reported drinking more in these circumstances than previously. As one participant noted:

> …I think we celebrated bigger, like all in celebrate. So, yeah, I would say that happened, like, because now there's always a reason to celebrate, which is cool, and we always are good-- we love a good celebration. But it felt like, ooh, we can really celebrate this time. We can be somewhere. We can do something about how we feel about this thing and so, yeah, we went in hard for the celebrations that were-- that we could do. So it was cool. It was cool to me because it's like, "Yeah, I get to celebrate and everybody is celebrating and we're all enjoying it." And I think it just kind of threw more into it because we couldn't celebrate for so long. And then now that I can celebrate, like, yeah, let's really get it in. (CM_01, 44-year-old Hispanic Black female with no AUD symptomology)

Another participant said

> So, here again, it's always about having a good time, the celebrating of something. You make up "Okay, we're going to celebrate the fact that I haven't seen you since last year," just the celebrating of the whole deal when you got together. (CM_03, 66-year-old non-Hispanic Black female with moderate AUD symptomology)

Whereas prior to the pandemic, celebratory drinking was described as related to significant milestones (e.g., getting a new job or getting married), in 2021 and later, spending time with loved ones, even casually, was described as an event to celebrate, and alcohol use during such gatherings was much more like (heavier) drinking patterns associated with special occasions, as opposed to light to moderate social drinking.

## Discussion

Results of this qualitative study suggest that participants' self-reported alcohol use during the COVID-19 pandemic was motivated by a confluence of both contextual and individual factors. Changes to social and environmental factors resulted in decreased barriers to alcohol use, increased alcohol availability and increased accessibility, all of which facilitated drinking. This finding corroborates quantitative studies suggesting that policies increasing the availability of alcohol during the pandemic were directly related to elevated drinking [47,48]. During the pandemic, social distancing measures were enacted [49] which vastly restricted what people could do and where they could go, resulting in many people being homebound. For some participants this confinement seems to have lessened barriers to alcohol use, that is, alcohol was seen as one of the few easily available options to reduce negative feelings and/or boredom. Furthermore, social distancing and quarantine measures effectively eliminated typical drinking contexts for many people.

Since social gatherings were not permitted and obligations that typically deter alcohol use, such as driving children to activities, were removed it became unclear when alcohol could or should be used. Concurrently, one participant believed that their peers used alcohol during the day (e.g., during work hours) and that such behavior was socially acceptable within the context of the pandemic. This participant described a lack of work obligations resulting in him and his colleagues drinking during their lunch breaks and sharing pictures of their beer with each other, socially reinforcing their shared change in alcohol use. In another instance, a participant described drinking because they and their friends had "nothing else to do". It could be inferred that these participants' alcohol use was socially reinforced by their peers. That is, the participant and their peers shared the experience of their alcohol use increasing due to boredom which was resultant of a reduction of other obligations. It is unclear why this social reinforcement occurred, but it could have been a strategy to mitigate feelings of shame around increased alcohol use. Presumably drinking during the day (e.g., during work hours) would not have been acceptable pre-pandemic. The participant and his colleagues who shared pictures of their beers may have done so as a way of validating their behavior- they may have been reassured that their behavior was not unusual given their peers were also drinking during the workday. This was not a common theme in the present analysis and should be explored in future work.

The confluence of free time and changes in descriptive and injunctive drinking norms resulted in some participants reporting using alcohol at times when they would not normally do so (e.g., during the workday). Such findings are corroborated by research showing that changes in descriptive norms around alcohol use (e.g., increase in perceived norms for drinks per week) was associated with self-reported increases in drinking during the pandemic [50,51]. Being secluded at home further facilitated alcohol use due to the removal of safety concerns (e.g., having to drive home while intoxicated) and accessibility of alcohol in the home.

Additional to the erasure of social barriers, the social isolation experienced by participants during the COVID-19 pandemic resulted in major shifts in participants' affective states. Participants described how throughout the pandemic they felt persistent negative emotions

including boredom and anxiety. Concurrently participants experienced a diminishment in the availability of positively emotionally valanced activities (e.g., socialization/recreation). The compounding impact of the simultaneous increase in negative affect and decreased access to reinforcing behaviors resulted in a perceptible shift in how alcohol was viewed and used among participants. During the pandemic alcohol became a "primary focus" where alcohol itself was viewed as a "treat" or something to look forward to. For some alcohol became an affective regulation strategy, where drinking was a means to both dispel negative emotions and increase positive feelings. This may help contextualize research demonstrating an association between depressive symptoms and alcohol use observed during the COVID-19 pandemic [52,53].

Changes in the availability and accessibility of alcohol during the COVID-19 pandemic further impacted self-reported alcohol use among participants. As participants described, several states saw an expansion of alcohol delivery policies. For example, in some states, liquor stores, restaurants and bars were allowed for the first time to deliver alcohol to people's residences [5,48]. This led to alcohol being easier to get and importantly did not require people to physically leave their homes. Evidence shows that people who said they used alcohol home delivery services during the pandemic consumed more alcohol and reported more drinking days and consumed larger volumes of alcohol at home than people who accessed alcohol by other means. As a result of this increased availability, participants described how their access to alcohol dramatically increased resulting in participants having alcohol continuously at home. This coupled with the social isolation and lack of typical alcohol use social cues likely further motivated increased alcohol use during the pandemic.

One participant mentioned how this increase in alcohol availability "softened" boundaries around alcohol use. This echoes statements made by other participants suggesting how changes in social context removed barriers to alcohol use. The increased ubiquity of alcohol seems to have resulted in a lowering of social constraints on alcohol use. That is, by increasing the availability of alcohol (i.e., to-go drinks, outdoor drinking venues) the participant perceived its use to be more socially acceptable. However, this sentiment was only shared by a single participant, so it is unclear the extent to which alcohol availability may have shifted social norms and thereby impacted alcohol use. Future research could further explore this topic.

Our results also suggest that the increases in alcohol use reported by participants continued after social distancing measures began to be lifted. As the restrictions originally imposed to curb the spread of COVID-19 eased, many participants described drinking alcohol to celebrate being able to (re)connect with loved ones and noted that social interactions that were previously part of everyday life felt like special occasions that deserved to be celebrated in the wake of long-term isolation. Alcohol use during such gatherings was described as more intense than pre-pandemic (i.e., participants engaged in heavy drinking). It can be inferred that this surge in alcohol use was due to participants wanting to enhance the positive experience of such events. It is also possible that this increased celebratory drinking may have been the result of their increased alcohol use during the pandemic (e.g., increased alcohol tolerance). These qualitative reports of desires for celebratory drinking upon easing of social distancing measures are consistent with per capita alcohol consumption estimates that continued to rise through 2021, where a 2.9% increase from 2020 was observed [10]. However, per capita alcohol consumption decreased in 2022, down 1.2% from 2021, suggesting that perhaps motivations to drink heavily to celebrate in-person social interactions did not persist, although more research is needed to establish the accuracy of this suggestion.

Our analyses focused on the contextual and individual level forces that motivated participants' self-reported increased alcohol use *during* the COVID-19 pandemic. It is currently

unknown what the long-term impacts of elevated pandemic-era alcohol use may be. It is plausible that this elevated substance use will have health consequences for many in the long-term. Indeed, some participants from our study discussed that they experienced health issues resultant from their increased use, including substance use disorder relapses, anxiety, and gastrointestinal distress. However, such accounts were anecdotal and did not reach saturation during thematic analysis. There is a need to further investigate the long-term impacts of increased alcohol use during the pandemic as we continue to move beyond the COVID-19 pandemic.

## Limitations

The present study has limitations which must be acknowledged when considering our findings. First, the present sample was derived from the NAS-C19 sample and was limited to those who reported drinking above NIAAA guidelines. As such, our findings may not be generalizable to the overall U.S. population. Although Wave 1 sampling procedures were designed to recruit a representative sample of the U.S. population, all participants were individuals who participated in all three waves of data collection and are likely different from the general population. In addition, our sample was majority non-Hispanic White, limiting our ability to draw inferences about how other racial and ethnic groups' alcohol use motivations and behaviors were impacted. This is especially important given that the Black Lives Matter movement occurred concurrently with the COVID-19 pandemic [54]. That is, the higher baseline stress described by our participants may have been even more severe among individuals from ethnic/racially minoritized groups who experienced both COVID-19 related stressors and increased saliency of racial marginalization. Research somewhat supports this claim, with evidence suggesting that alcohol use increased among individuals from ethnic/racially minoritized groups during the pandemic [14,55]. In addition, just over half of our participants self-identified as essential workers, but we did not collect data on participants' specific occupations. As such we can only make limited inferences on how alcohol use may have been impacted among different types of essential workers (e.g., healthcare workers). However, regardless of occupation, it is likely given the high levels of stress and psychological trauma reported by essential workers [56] that their alcohol use was impacted by the COVID-19 pandemic [57].

The present work is also limited due to a lack of saturation of several topics germane to alcohol use during the COVID-19 pandemic. For example, one participant described how their reported alcohol use may have been a way for them to exert control during a time when their autonomy was limited. This was only expressed by a single participant, but such behavior may have been experienced by others. Similarly, one participant implied that increased alcohol availability impacted perceived social norms around alcohol use. However, this was the only time this topic arose-- we have no other analogous participant statements in our dataset. Since this topic never reached saturation, we cannot draw conclusions about how social forces and alcohol availability interacted to impact behavior. Further research is needed to explore these topics.

Another limitation is that our ability to make inferences about potential mechanisms of or reasons for changes in alcohol use is limited by what was salient to participants at the time of the interviews. For example, besides the expansion of alcohol delivery services (e.g., to-go drinks) there were likely other factors which changed availability (e.g., increased demand for alcohol during the pandemic). However, since participants did not discuss other changes in alcohol availability, we can only guess as to what those were. In a similar way, although some participants discussed their employment arrangements during COVID-19 and its association with their alcohol use, this was not a focus of the interviews. As such we have limited data on

participants' COVID-19 work arrangements and how such arrangements may have impacted their alcohol use.

It is also important to note that the present study, like all retrospective research, relied on participants to provide post hoc explanations for their behavior. Relying on retrospective data presents certain limitations which impacts what can be inferred from the data due to a lack of accuracy. Research suggests that alcohol use self-report accuracy diminishes over time [58,59] and that retrospective estimates of substance use may be more representative of participants' current use at the time of assessment rather than of the time in question [60,61]. There is also evidence that the accuracy of self-reported alcohol use diminishes when participants engage in heavy/binge alcohol use [62]. In addition, other research suggests that participants' current emotional states [63] and social desirability [64] influences retrospective reporting of past events (e.g., alcohol use). Therefore, participants attributing their increased alcohol use to negative affect may be inaccurate. Taken together, this evidence suggests that the accuracy of participants' accounts of alcohol use and their potential motivations to use alcohol may have been inaccurate. However, it should be noted that all participants in the current study reported drinking above NIAAA guidelines during wave 2 of the NAS-C19 study. That is, these participants' retrospective accounts of increased alcohol use during the interviews were corroborated by their own data collected from during the pandemic. Given that the NAS-C19 survey was also retrospective (it asked about alcohol use over the past several months) it is still possible that these data may lack accuracy. Nevertheless, this corroboration increases our confidence that at the very least participants' recollections about the changes in the general patterns of alcohol use during the COVID-19 pandemic were accurate.

A second limitation of relying on retrospective reports is it precludes us from making comparisons across time points. That is, our qualitative interviews are cross-sectional retrospective data, therefore this makes it challenging to discern how behaviors differed before and during the pandemic. Although we explicitly asked participants to describe how they perceived their alcohol use motivations were impacted by the pandemic, as described above, said reports may have been inaccurate. Given we do not have a means to more objectively compare drinking motivation across time points limits our ability to confidently state how the pandemic impacted alcohol use motivation.

## Conclusions

The current study provides evidence for how the COVID-19 pandemic may have impacted motives to use alcohol among a sample of U.S.-based alcohol using adults. Through qualitative analysis of participants' accounts we determined that both contextual and individual level forces interacted to shape participants' self-reported alcohol use during this time. Contextual factors including reduced barriers to alcohol use (e.g., changes in work schedules), and increases in both alcohol availability (e.g., legalization of alcohol delivery) and accessibility (e.g., being homebound where alcohol was present) facilitated elevated drinking during the pandemic. These factors helped catalyze participants' individual reasons for drinking including drinking as a means of emotional regulation (e.g., to manage chronic boredom resultant of being confined to their home). These findings help to frame quantitative research showing elevated alcohol use during the COVID-19 pandemic. To date research has focused on either environmental (e.g., expansion of alcohol home delivery; [5,48]) or individual (e.g., negative affect; [65]) level factors' impact on pandemic era drinking separately. Our research contributes to the field by examining these concurrently and explored how these forces interacted together to influence participants' motivations to use alcohol in a global pandemic. This adds to the field of public health and offers insights into how to mitigate future increases in

substance use post-disasters. Findings suggest it is inadvisable to increase alcohol availability and accessibility in a global crisis rife with fear, boredom and isolation if increases in alcohol use and related harms are to be averted. Further, given the potential for changes in social norms around alcohol use during a global crisis as we observed here, social norms campaigns [66] leveraging normative messaging highlighting that not everyone is engaging in increased or heavy alcohol use may help reduce the impact of more permissive drinking norms. Future research is needed to investigate the long-term impacts of pandemic era changes in alcohol use, and it is vitally important for future investigations to examine how such impacts may manifest differently across different subgroups of the U.S. population.

## Supporting Information

**S1 Table. Summary of identified themes and exemplar quotes.**
(DOCX)

Acknowledgements: We, the authors, acknowledge and express gratitude to our participants for volunteering their time to participate in our research. We also acknowledge our colleagues Deidre Patterson, Yu Ye and Drs. William C. Kerr, Thomas K. Greenfield, and Libo Li for their work on the National Alcohol Survey COVID Cohort.

## Author contributions

**Conceptualization:** Cat Munroe.

**Data curation:** Anthony Surace.

**Formal analysis:** Anthony Surace, Priscilla Martinez, Cat Munroe.

**Funding acquisition:** Priscilla Martinez.

**Investigation:** Priscilla Martinez, Cat Munroe.

**Methodology:** Anthony Surace, Cat Munroe.

**Project administration:** Anthony Surace.

**Resources:** Priscilla Martinez.

**Software:** Priscilla Martinez.

**Supervision:** Priscilla Martinez.

**Writing – original draft:** Anthony Surace, Cat Munroe.

**Writing – review & editing:** Anthony Surace, Priscilla Martinez, Cat Munroe.

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
