## [Decision Letter · Decision Letter 0]

9 Oct 2024

Dear Dr. Surace,

Thank you for submitting your manuscript to PLOS ONE. After careful consideration, we feel that it has merit but does not fully meet PLOS ONE’s publication criteria as it currently stands. Therefore, we invite you to submit a revised version of the manuscript that addresses the points raised during the review process.

We look forward to receiving your revised manuscript.

Kind regards,

Colin Jerolmack, PhD

Academic Editor

PLOS ONE

Journal Requirements:

2. Please note that funding information should not appear in the Acknowledgments section or other areas of your manuscript. We will only publish funding information present in the Funding Statement section of the online submission form. Please remove any funding-related text from the manuscript. 

3. In the online submission form, you indicated that “Data are available upon request.”

3. Uploaded as supplementary information.

4. We note you have included a table to which you do not refer in the text of your manuscript. Please ensure that you refer to Table 2 in your text; if accepted, production will need this reference to link the reader to the Table.

**Additional Editor Comments:**

Dear Dr. Surace,

First of all, my sincere apologies for how long it has taken us to reach a decision on your manuscript. I had an an unusually difficult time securing reviewers, many of whom either never responded to the request or agreed but didn't turn in a report. But I am pleased to report that I did secure two thorough reviews from two scholars who are well placed to judge the contributions of this manuscript, especially on methodological grounds. As you know, PLOS ONE pays particular attention to methodological rigor, and both reviewers are accomplished qualitative researchers who also have written about social aspects of either COVID or health. Both see merit in the paper and suggest a revise and resubmit. I concur. I will try to send your revision back to the same reviewers but may need to secure new ones if they are unavailable.

The reports speak for themselves and don't contradict each other, so i don't feel the need to provide in depth comments of my own. Both reviewers, while sympathetic to the enterprise of interviewing a relatively small and non-random sample of people, encourage you to revise your claims about how this approach is superior to other approaches that are restricted to "specific populations," as your own sample is undoubtedly a specific population. Providing more details about the attributes and other background characteristics of your participants, as spelled out by R2, would also be useful. regarding the framing and contribution, both reviewers think you could and should do more to situate your findings and clarify their contribution. For R2, that's analyzing how contextual and individual factors interact to lead to changes in perception of social acceptability of drinking, for R1, that's saying a bit more about related research on the pandemic and social isolation. Both reviewers also raise some questions about how you analyze your data--e.g., did subjects drink a lot at home before the pandemic, and it's just a matter that they are home more, or did something change that made them START drinking at home? DO you habe more data that illustrates social isolation, which is a big point in the discussion but not prominent in the data?

I encourage you to read the reviews closely, and I look forward to seeing a revised version of this paper.

Colin Jerolmack

Reviewers' comments:

Reviewer's Responses to Questions

**Comments to the Author**

1. Is the manuscript technically sound, and do the data support the conclusions?

Reviewer #1: Yes

Reviewer #2: Yes

2. Has the statistical analysis been performed appropriately and rigorously?

Reviewer #1: N/A

Reviewer #2: N/A

3. Have the authors made all data underlying the findings in their manuscript fully available?

Reviewer #1: Yes

Reviewer #2: Yes

4. Is the manuscript presented in an intelligible fashion and written in standard English?

Reviewer #1: Yes

Reviewer #2: Yes

Reviewer #1: Thank you for sharing this manuscript. The main claims of the paper are that contextual and individual level factors such as reductions in barriers to alcohol use, increases in alcohol availability, and increases in alcohol accessibility contributed to increased alcohol use during the COVID-19 pandemic. These environmental factors interacted with personal emotions to increase consumption. These claims are significant given the limitations of quantitative investigations of increased alcohol use, which have not addressed motives and rationales.

The main strengths of the paper are its methodological rigor, contextual focus, and rich data. The qualitative approach is appropriate for exploring in-depth rationales, motivations, practices, and beliefs that led to increased alcohol use. The analysis effectively captured contextual factors unique to the pandemic which provides insights into the complex interplay between environmental conditions and individual behavior. The detailed quotes from participants offer vivid illustrations of their experiences.

The main limitations of the paper are incomplete analyses of the data, lack of theoretical distinction between the environmental factors, and an insufficient literature review. See below for details on specific areas for improvement:

1. The literature review could be developed, especially as it pertains to overall public health or social science literature published during the pandemic. It could start broad and then delve into the particular issue of alcohol use. The literature review is also lacking in historical context. Comparisons could be made to previous crises or emergencies, which would provide a broader perspective on how the pandemic’s impact on alcohol use compares to other major events. Additionally, the literature review could address social isolation and social reinforcement which seem to be key to the analysis.

2. Lines 90-91 mention that the limits of other qualitative research is that it is restricted to specific populations, but your population is restricted as well (it is not representative). I think this should be stated more explicitly.

3. Lines 110-119 mention surveys conducted in the past. This manuscript could draw out the differences between the surveys done and your interviews a bit more.

4. The methods section does not state a response rate.

5. Lines 190-192 restate the three environmental factors which contributed to increased alcohol use (reduction in barriers, increased availability, and increased accessibility). It is not immediately clear to me how different these factors are. Accessibility and availability of alcohol increase seem particularly similar to me. Can you draw out the differences?

6. One of your arguments is that participants were more likely to drink a lot at home versus at bars. Participants likely drank at home before the pandemic as well, was it always more at home? Generally, I am trying to understand what is different about being at home during a pandemic versus these participants being at home for other reasons. When they had long weekends or breaks from work did the same patterns in alcohol use arise? I’m not sure how “always having beer in the house” or “having alcohol in their homes” during the pandemic was different than the alcohol in the house beforehand? What’s special about the pandemic here?

7. Lines 253-256 mentions “socially reinforcing their shared change in alcohol use.” Why might this have happened? Did others experience a similar reinforcement? Why were there a lack of work obligations for people who didn’t lose their job? Why were there greater workloads on-site, if that was the case?

8. Besides delivery services and to-go drinks, were there other things that changed availability? You also mention this availability lowering social constraints/making it more socially acceptable but don’t flesh out an argument for this. There is a lot more that could be said about social pressures and shifting norms, and the mechanisms by which it became more socially acceptable.

9. The point the interviewee makes about relaxation in lines 370-377 could be fleshed out more. Seems an important argument about the stress of the pandemic creating a desire for relaxation mechanisms. So it might not just be accessibility of alcohol making it an “attractive option for stress relief,” but the pandemic itself (I know you go more in depth on this in theme 4, so this quote could be moved to that section)

10. For theme 4, can you spell out the negative emotions? You jump into stress and boredom right away, but it is unclear what the range of emotions described by participants are.

11. The ideas about control are really interesting. The manuscript could emphasize this and analyze the quotes further. Control seems qualitatively different than boredom, for example, and could use its own analysis.

12. Theme 5 is too short relative to the other sections.

13. Social isolation comes up in the discussion but not as much in the results. Any more quotes or narratives that highlight this social isolation? Also, there is a lot of literature on this that could be included in the literature review.

14. In the limitations section, it would be helpful to mention what kind of workers you interviewed. It seems work arrangements are an important aspect to bring up as part of the analysis. How did their work situations uniquely contribute to your findings? The demographics table could include work arrangements as well.

I enjoyed reading this paper and I think it makes important contributions to our understandings of alcohol use.

Reviewer #2: This article analyzes the contextual and individual-level forces that motivated increased alcohol use during the first year of the COVID-19 pandemic. Using qualitative interviews with 26 participants, it describes how changes in the availability and accessibility of alcohol interacted with social isolation, negative emotions, and lack of routines to make alcohol a more recurrent and acceptable coping mechanism.

The paper is clear, well written, and adequate in its use of evidence to support its claims. My comments for revision are related to (1) strengthening the statement of motivation for the study; (2) providing more details on the sample and how they affect the paper’s results, (3) clarifying the contributions.

1. The article currently justifies its methods and approach by stating that there is little qualitative research on alcohol consumption during the COVID pandemic and that the existing qualitative research is limited due to its “focus on specific populations.” Yet this article hardly remedies that, since its sample is small (n=26) and comprises a very specific population as it’s derived from the NAS-C19 survey, thus limited to people who have reported drinking above NIAAA guidelines. Plus, the sample is heavily made of White, college-educated participants. I am not suggesting in any way that this compromises the article’s results, but to motivate the paper based on the sampling limitations of previous qualitative research while exhibiting these same kinds of limitations is not the best approach.

Likewise, to say that “Another limitation of qualitative work to date has been its use of open-ended survey responses to assess motives for drinking which severely limits the amount and depth of qualitative data” (lines 97-98) strikes me as odd, since survey studies are not qualitative.

2. There are several demographic characteristics in the sample that either need to be expanded on or added to the table. Among the ones I suggest expanding on are:

a. Employment status: at several points in the article we read about work-related obligations or lack thereof, or unemployment, impacting alcohol consumption (i.e. line 238). How exactly does this variable mediate the patterns the authors find? Did any of these people lose their jobs during the pandemic?

b. Income: from Table 1 we learn that about half of the sample earns: $70,000 or more. Is this individual or household income? What about the rest?

c. Did people with Alcohol Use Disorder Symptoms (57.7% of the sample) differ from the rest in the patterns identified?

Two characteristics that are not included in Table 1 and I suggest including:

a. Children. Several times we read about children in participants’ quotes (line 207; 246-247; 410) and it seems like both presence and absence of children act as motivations for drinking. I suggest working this into the analysis.

b. Geographic location. We know nothing about where participants live, which seems important because some contextual factors could be shaped by different local policies (i.e., stay-at-home orders; allowing bars to remain open; alcohol delivery).

3. Perhaps related to my first point about the motivation for this article, I’d suggest highlighting its contributions a bit more. For instance, you could comment on whether previous studies differentiate between contextual and individual level factors in participants’ alcohol use. Or, and this might be beyond the scope of this analysis, you could do more with a theme I find really interesting which is how the interaction of contextual and individual factors lead to a change in perceptions of social acceptability in alcohol use. In the discussion (lines 520-523) we read that the lack of usual routines (work, school) made it “unclear when alcohol could or should be used.” It seems like this is a novel contribution of this study that could be highlighted.

**Do you want your identity to be public for this peer review?** For information about this choice, including consent withdrawal, please see our Privacy Policy

Reviewer #1: **Yes: ** Michelle Cera

Reviewer #2: **Yes: ** Guillermina Altomonte

---

## [Author Response · Author response to Decision Letter 1]

6 Dec 2024

Reviewer #1: Thank you for sharing this manuscript. The main claims of the paper are that contextual and individual level factors such as reductions in barriers to alcohol use, increases in alcohol availability, and increases in alcohol accessibility contributed to increased alcohol use during the COVID-19 pandemic. These environmental factors interacted with personal emotions to increase consumption. These claims are significant given the limitations of quantitative investigations of increased alcohol use, which have not addressed motives and rationales.

The main strengths of the paper are its methodological rigor, contextual focus, and rich data. The qualitative approach is appropriate for exploring in-depth rationales, motivations, practices, and beliefs that led to increased alcohol use. The analysis effectively captured contextual factors unique to the pandemic which provides insights into the complex interplay between environmental conditions and individual behavior. The detailed quotes from participants offer vivid illustrations of their experiences.

The main limitations of the paper are incomplete analyses of the data, lack of theoretical distinction between the environmental factors, and an insufficient literature review. See below for details on specific areas for improvement:

1. The literature review could be developed, especially as it pertains to overall public health or social science literature published during the pandemic. It could start broad and then delve into the particular issue of alcohol use. The literature review is also lacking in historical context. Comparisons could be made to previous crises or emergencies, which would provide a broader perspective on how the pandemic’s impact on alcohol use compares to other major events. Additionally, the literature review could address social isolation and social reinforcement which seem to be key to the analysis.

Response: We thank the reviewer for this comment. We have added text reviewing the literature on changes in alcohol use post-disasters to our introduction as well as social isolation in the context of COVID-19. This includes a comparison of how COVID-19 may have impacted alcohol use in different ways due to its unique nature (see lines 68-84).

2. Lines 90-91 mention that the limits of other qualitative research is that it is restricted to specific populations, but your population is restricted as well (it is not representative). I think this should be stated more explicitly.

Response: We agree with the reviewer- our sample was quite specific. We have edited the manuscript to note this and have removed this limitation from our description of past literature (see lines 105-122).

3. Lines 110-119 mention surveys conducted in the past. This manuscript could draw out the differences between the surveys done and your interviews a bit more.

Response: We have added a description outlining the differences of the previously cited survey studies to the Introduction (see lines 123-136).

4. The methods section does not state a response rate.

Response: We have included the response rate for the qualitative interviews in the Methods section (see line 162).

5. Lines 190-192 restate the three environmental factors which contributed to increased alcohol use (reduction in barriers, increased availability, and increased accessibility). It is not immediately clear to me how different these factors are. Accessibility and availability of alcohol increase seem particularly similar to me. Can you draw out the differences?

Response: We agree that the difference between availability and accessibility can be difficult to distinguish and the importance of having these clearly defined and differentiated. We have more clearly defined these themes on lines 216-222.

6. One of your arguments is that participants were more likely to drink a lot at home versus at bars. Participants likely drank at home before the pandemic as well, was it always more at home? Generally, I am trying to understand what is different about being at home during a pandemic versus these participants being at home for other reasons. When they had long weekends or breaks from work did the same patterns in alcohol use arise? I’m not sure how “always having beer in the house” or “having alcohol in their homes” during the pandemic was different than the alcohol in the house beforehand? What’s special about the pandemic here?

Response: We agree with the reviewer that more information is needed to distinguish between participants’ drinking behavior pre- and during the pandemic. We have created an additional table (“Table 2 Participant Alcohol Use”) which highlights how participants’ drinking changed, including the frequency of self-reported alcohol use while at home (which increased during the pandemic). However, it is beyond the scope of our data to determine how extended periods at home (e.g., “long weekends”) pre-pandemic may have influenced alcohol use.

7. Lines 253-256 mentions “socially reinforcing their shared change in alcohol use.” Why might this have happened? Did others experience a similar reinforcement? Why were there a lack of work obligations for people who didn’t lose their job? Why were there greater workloads on-site, if that was the case?

Response: We thank the reviewer for these questions- these are important things to consider when analyzing the data. We expanded the Discussion section to include a brief discussion of potential reasons for this social reinforcement (see lines 618-632). Unfortunately, we did not ask participants about the specifics of their work obligations, so it is beyond the scope of these data to address potential differences in employment responsibilities and alcohol use.

8. Besides delivery services and to-go drinks, were there other things that changed availability? You also mention this availability lowering social constraints/making it more socially acceptable but don’t flesh out an argument for this. There is a lot more that could be said about social pressures and shifting norms, and the mechanisms by which it became more socially acceptable.

Response: Reviewer 1 brings up some interesting points with this comment. We agree that there were likely other forces driving changes in alcohol availability during the pandemic, However, we are limited by our data- alcohol delivery and to go drinks were the examples of increased alcohol availability shared by participants during our interviews. They did not share other ways that availability may have been increased. Similarly, only one participant implied that the increased availability of alcohol may have impacted social acceptability of alcohol use. Given this theme did not reach saturation, we feel it would be inappropriate to speculate on the potential impact of social pressures/norms of alcohol use in this manuscript. We acknowledge these limitations and advocate for further work to be done in these topic areas (see lines 726-745).

9. The point the interviewee makes about relaxation in lines 370-377 could be fleshed out more. Seems an important argument about the stress of the pandemic creating a desire for relaxation mechanisms. So it might not just be accessibility of alcohol making it an “attractive option for stress relief,” but the pandemic itself (I know you go more in depth on this in theme 4, so this quote could be moved to that section)

Response: We agree with the reviewer and have moved this quote and discussion to the theme four section (see lines 437-454).

10. For theme 4, can you spell out the negative emotions? You jump into stress and boredom right away, but it is unclear what the range of emotions described by participants are.

Response: We have spelled out the negative emotions expressed by participants on lines 432-434.

11. The ideas about control are really interesting. The manuscript could emphasize this and analyze the quotes further. Control seems qualitatively different than boredom, for example, and could use its own analysis.

Response: We agree that this is a fascinating topic. We reviewed the interview transcripts and unfortunately this was the only instance where this topic emerged. In hindsight we should have asked participants about this, but it was not something we had considered when we were conducting these interviews. We acknowledge this limitation in the manuscript see lines 727-730.

12. Theme 5 is too short relative to the other sections.

Response: We have expanded the section on theme 5 including a more detailed description of participants’ use of alcohol (see lines 530-566).

13. Social isolation comes up in the discussion but not as much in the results. Any more quotes or narratives that highlight this social isolation? Also, there is a lot of literature on this that could be included in the literature review.

Response: We have expanded the theme 4 results to include quotes relating to social isolation among participants and its impact on their alcohol use (see lines 499-529)

14. In the limitations section, it would be helpful to mention what kind of workers you interviewed. It seems work arrangements are an important aspect to bring up as part of the analysis. How did their work situations uniquely contribute to your findings? The demographics table could include work arrangements as well.

Response: We agree with the reviewer that participants’ employment likely impacted their behaviors during the COVID-19 pandemic. Unfortunately, we have limited data regarding their specific jobs and work arrangements- although some participants mentioned their occupation it was not a standard survey/interview question. We are therefore limited in our ability to offer detailed descriptions of participants’ jobs/work arrangements. We have expanded the demographics included in table 1 to include more detailed employment information (e.g., the number of participants who self-identified as essential worders during COVID). We have also expanded the limitations section to reflect this (see lines 726-745)

I enjoyed reading this paper and I think it makes important contributions to our understandings of alcohol use.

Response: We thank the reviewer for this positive comment and for the constructive critiques that have helped to improve the quality of the manuscript.

Reviewer #2: This article analyzes the contextual and individual-level forces that motivated increased alcohol use during the first year of the COVID-19 pandemic. Using qualitative interviews with 26 participants, it describes how changes in the availability and accessibility of alcohol interacted with social isolation, negative emotions, and lack of routines to make alcohol a more recurrent and acceptable coping mechanism.

The paper is clear, well written, and adequate in its use of evidence to support its claims. My comments for revision are related to (1) strengthening the statement of motivation for the study; (2) providing more details on the sample and how they affect the paper’s results, (3) clarifying the contributions.

1. The article currently justifies its methods and approach by stating that there is little qualitative research on alcohol consumption during the COVID pandemic and that the existing qualitative research is limited due to its “focus on specific populations.” Yet this article hardly remedies that, since its sample is small (n=26) and comprises a very specific population as it’s derived from the NAS-C19 survey, thus limited to people who have reported drinking above NIAA guidelines. Plus, the sample is heavily made of White, college-educated participants. I am not suggesting in any way that this compromises the article’s results, but to motivate the paper based on the sampling limitations of previous qualitative research while exhibiting these same kinds of limitations is not the best approach. Likewise, to say that “Another limitation of qualitative work to date has been its use of open-ended survey responses to assess motives for drinking which severely limits the amount and depth of qualitative data” (lines 97-98) strikes me as odd, since survey studies are not qualitative.

Response: We agree with the reviewer. In hindsight, our sample was quite specific. We have edited the manuscript and have removed this limitation from our description of past literature. We also agree that survey studies are not qualitative. The cited studies treated responses to open-ended survey questions as qualitative data. We added further details about why this is a limitation of the literature (see lines 123-136).

There are several demographic characteristics in the sample that either need to be expanded on or added to the table. Among the ones I suggest expanding on are:

a. Employment status: at several points in the article we read about work-related obligations or lack thereof, or unemployment, impacting alcohol consumption (i.e. line 238). How exactly does this variable mediate the patterns the authors find? Did any of these people lose their jobs during the pandemic?

Response: We agree with reviewer 2 that employment status is an important factor in these analyses. We have expanded table 1 to include more details about participants’ employment (e.g., if they became unemployed during the pandemic. We note, however, that it is beyond the scope of these qualitative data to conduct mediation analysis.

b. Income: from Table 1 we learn that about half of the sample earns: $70,000 or more. Is this individual or household income? What about the rest?

Response: We have clarified in table 1 that this is referring to household income. $70,000 was the median household income for the sample.

c. Did people with Alcohol Use Disorder Symptoms (57.7% of the sample) differ from the rest in the patterns identified?

Response: We did not observe notable differences in our qualitative analyses (i.e., theme frequency) based on participants’ AUD symptomology scores. We have now included categorical AUD severity scores to participant information post quotes.

Two characteristics that are not included in Table 1 and I suggest including:

a. Children. Several times we read about children in participants’ quotes (line 207; 246-247; 410) and it seems like both presence and absence of children act as motivations for drinking. I suggest working this into the analysis.

Response: We agree with reviewer 2 that given childcare came up during the interviews that that the number if minors living in participants’ households is important information to include. We have therefore added it to table 1.

b. Geographic location. We know nothing about where participants live, which seems important because some contextual factors could be shaped by different local policies (i.e., stay-at-home orders; allowing bars to remain open; alcohol delivery).

Response: We agree with reviewer 2 that contextual factors could shape alcohol policies. We have therefore added geographical location to table 1.

3. Perhaps related to my first point about the motivation for this article, I’d suggest highlighting its contributions a bit more. For instance, you could comment on whether previous studies differentiate between contextual and individual level factors in participants’ alcohol use. Or, and this might be beyond the scope of this analysis, you could do more with a theme I find really interesting which is how the interaction of contextual and individual factors lead to a change in perceptions of social acceptability in alcohol use. In the discussion (lines 520-523) it “unclear when alcohol could or should be used.” It seems like this is a novel contribution of this study that could be highlighted.

Response: We have expanded our Discussion section to highlight our contributions to the field more see lines 770-772. We agree with reviewer 2 that the interaction of contextual (e.g., alcohol availability) and individual factors (e.g., perceived social norms) may have impacted alcohol use. However, only one participant described this phenomenon. Given this lack of saturation of the theme, it is beyond the scope of the data to speculate on the potential impact of social pressures/norms of alcohol use in this manuscript. We acknowledge these limitations and advocate for fur

---

## [Decision Letter · Decision Letter 1]

3 Jan 2025

Dear Dr. Surace,

Thank you for submitting your manuscript to PLOS ONE. After careful consideration, we feel that it has merit but does not fully meet PLOS ONE’s publication criteria as it currently stands. Therefore, we invite you to submit a revised version of the manuscript that addresses the points raised during the review process.

We look forward to receiving your revised manuscript.

Kind regards,

Colin Jerolmack, PhD

Academic Editor

PLOS ONE

Journal Requirements:

Additional Editor Comments:

Hello and happy new year--an appropriate moment to write given the topic of your paper and the prevalence of drinking at this time of year!

Thank you for your careful revisions. I am prepared to accept the paper after you make the following minor revisions:

1. The reviewer points out a small discrepancy between reporting and sample; please reconcile

2. I think you need to say a bit more about the potential problems of self-reporting alcohol use. I appreciate that you get at some of the issues in the "limitations" portion of the paper, but the influence of affective states on reporting of alcohol isn't the only potential issue. We can imagine how selective or inaccurate recall may play a role, or social desirability bias, etc. And the issue of recall presumably grows the longer in time you ask people to look back upon--e.g., a survey asking how many drinks you had YESTERDAY is likely more accurate than one asking in the past WEEK, which in turn is better than last MONTH, and so on. I understand you are more interested in patterns [drinking "more"] than specific numbers of drinks, but the same kind of issues may apply. There must be some studies that try to gauge the accuracy of self-reporting on alcohol consumption that can help us understand how concerned we should be, no?

3. Relatedly, and given that these are accounts, you need to go through the entire ms. and add qualifiers such as "reported" or "said" in front of descriptions of actions if they are based on self reports. For instance, "evidences shows that people who SAID THEY used more alcohol."

4. Also related to the issue of accounts, I would like you to be more careful about talking about MOTIVATION. I understand that this is the way people talk in this field, but it's very hard to ever get at motivation and is made all the harder when one is using one-off interviews in which people are asked to look back. Can we really say that we have captured motivation when we have no T1 to compare to T2? What we actually have are cross-sectional data [the one shot interview] in which you are relying on accounts to construct a narrative in which you can say that event or mood at T1 caused behavior at T2. I leave it up to you how to deal with it, but I advise being more careful about confidently stating claims about motivation.

Reviewers' comments:

Reviewer's Responses to Questions

**Comments to the Author**

Reviewer #1: All comments have been addressed

Reviewer #2: All comments have been addressed

2. Is the manuscript technically sound, and do the data support the conclusions?

Reviewer #1: Yes

Reviewer #2: Yes

3. Has the statistical analysis been performed appropriately and rigorously?

Reviewer #1: Yes

Reviewer #2: N/A

4. Have the authors made all data underlying the findings in their manuscript fully available?

Reviewer #1: Yes

Reviewer #2: No

5. Is the manuscript presented in an intelligible fashion and written in standard English?

Reviewer #1: Yes

Reviewer #2: Yes

Reviewer #1: (No Response)

Reviewer #2: Thank you for addressing my comments. My one observation for this revised version is that while two respondents mention the presence of children (PM_11 and PM_08), there is only one respondent indicated as having children in the household in Table 1.

**Do you want your identity to be public for this peer review?** For information about this choice, including consent withdrawal, please see our Privacy Policy

Reviewer #1: **Yes: ** Michelle Cera

Reviewer #2: **Yes: ** Guillermina Altomonte

---

## [Author Response · Author response to Decision Letter 2]

27 Jan 2025

1. The reviewer points out a small discrepancy between reporting and sample; please reconcile

Response: We thank the reviewer for pointing this out. The number we had previously added was the average number of children in the household (not the frequency of participants with children in their household). We have edited table 1 to make this more explicit.

2. I think you need to say a bit more about the potential problems of self-reporting alcohol use. I appreciate that you get at some of the issues in the "limitations" portion of the paper, but the influence of affective states on reporting of alcohol isn’t the only potential issue. We can imagine how selective or inaccurate recall may play a role, or social desirability bias, etc. And the issue of recall presumably grows the longer in time you ask people to look back upon--e.g., a survey asking how many drinks you had YESTERDAY is likely more accurate than one asking in the past WEEK, which in turn is better than last MONTH, and so on. I understand you are more interested in patterns [drinking "more"] than specific numbers of drinks, but the same kind of issues may apply. There must be some studies that try to gauge the accuracy of self-reporting on alcohol consumption that can help us understand how concerned we should be, no?

Response: We thank the editor for their feedback. We agree that retrospective self-report methodologies have inherent limitations including recall biases which can impact accuracy of data collected. We have expanded the limitations section to acknowledge this. See lines 747-775.

3. Relatedly, and given that these are accounts, you need to go through the entire ms. and add qualifiers such as “reported" or "said" in front of descriptions of actions if they are based on self reports. For instance, "evidences shows that people who SAID THEY used more alcohol."

Response: We thank the editor for pointing this out. We have added qualifiers to description of alcohol use in the manuscript specifying that these reflect self-reported behaviors. For example, see lines 294, 401, and 601.

4. Also related to the issue of accounts, I would like you to be more careful about talking about MOTIVATION. I understand that this is the way people talk in this field, but it's very hard to ever get at motivation and is made all the harder when one is using one-off interviews in which people are asked to look back. Can we really say that we have captured motivation when we have no T1 to compare to T2? What we actually have are cross-sectional data [the one shot interview] in which you are relying on accounts to construct a narrative in which you can say that event or mood at T1 caused behavior at T2. I leave it up to you how to deal with it, but I advise being more careful about confidently stating claims about motivation.

Response: We thank the editor for their feedback. We have expanded the limitations section of the manuscript to reflect this feedback (see lines 773-78). We have also hedged some of our previous statements regarding alcohol use motivations (e.g., lines 30, 601 and 780).

---

## [Editor Report · Decision Letter 2]

12 Feb 2025

“My drinking was way worse during the pandemic”: A Qualitative Analysis of Contextual and Individual Factors Impacting Alcohol Use During the COVID-19 Pandemic

PONE-D-24-29555R2

Dear Dr.Surace,

We’re pleased to inform you that your manuscript has been judged scientifically suitable for publication and will be formally accepted for publication once it meets all outstanding technical requirements.

Kind regards,

Colin Jerolmack, PhD

Academic Editor

PLOS ONE
---

## [Editor Report · Acceptance letter]

PONE-D-24-29555R2

PLOS ONE

Dear Dr. Surace,

I'm pleased to inform you that your manuscript has been deemed suitable for publication in PLOS ONE. Congratulations! Your manuscript is now being handed over to our production team.

Kind regards,

on behalf of

Dr. Colin Jerolmack

Academic Editor

PLOS ONE